# *Chlamydia Psittaci* ST24: Clonal Strains of One Health Importance Dominate in Australian Horse, Bird and Human Infections

**DOI:** 10.3390/pathogens10081015

**Published:** 2021-08-11

**Authors:** Susan I. Anstey, Vasilli Kasimov, Cheryl Jenkins, Alistair Legione, Joanne Devlin, Jemima Amery-Gale, James Gilkerson, Sam Hair, Nigel Perkins, Alison J. Peel, Nicole Borel, Yvonne Pannekoek, Anne-Lise Chaber, Lucy Woolford, Peter Timms, Martina Jelocnik

**Affiliations:** 1Genecology Research Centre, University of the Sunshine Coast, Sippy Downs, QLD 4557, Australia; susan.anstey@research.usc.edu.au (S.I.A.); vasilli.kasimov@research.usc.edu.au (V.K.); ptimms@usc.edu.au (P.T.); 2NSW Department of Primary Industries, Elizabeth Macarthur Agricultural Institute, Menangle, NSW 2568, Australia; cheryl.jenkins@dpi.nsw.gov.au; 3Asia Pacific Centre for Animal Health, Melbourne Veterinary School, The University of Melbourne, Parkville, VIC 3010, Australia; legionea@unimelb.edu.au (A.L.); devlinj@unimelb.edu.au (J.D.); j.amery-gale@student.unimelb.edu.au (J.A.-G.); jrgilk@unimelb.edu.au (J.G.); 4WA Department of Primary Industries and Regional Development, South Perth, WA 6151, Australia; sam.hair@dpird.wa.gov.au; 5School of Veterinary Science, The University of Queensland, Gatton, QLD 4343, Australia; hosvetsci@uq.edu.au; 6Centre for Planetary Health and Food Security, Griffith University, Nathan, QLD 4111, Australia; a.peel@griffith.edu.au; 7Institute of Veterinary Pathology, Vetsuisse Faculty, University of Zurich, 8066 Zurich, Switzerland; nicole.borel@uzh.ch; 8Department of Medical Microbiology, Amsterdam UMC, University of Amsterdam, 3508 Amsterdam, The Netherlands; y.pannekoek@amsterdamumc.nl; 9School of Animal and Veterinary Sciences, The University of Adelaide, Roseworthy, SA 5371, Australia; anne-lise.chaber@adelaide.edu.au (A.-L.C.); lucy.woolford@adelaide.edu.au (L.W.)

**Keywords:** *Chlamydia psittaci*, genetic diversity, Australia, MLST, novel strains, novel hosts, *omp*A genotyping

## Abstract

*Chlamydia psittaci* is traditionally regarded as a globally distributed avian pathogen that can cause zoonotic spill-over. Molecular research has identified an extended global host range and significant genetic diversity. However, Australia has reported a reduced host range (avian, horse, and human) with a dominance of clonal strains, denoted ST24. To better understand the widespread of this strain type in Australia, multilocus sequence typing (MLST) and *omp*A genotyping were applied on samples from a range of hosts (avian, equine, marsupial, and bovine) from Australia. MLST confirms that clonal ST24 strains dominate infections of Australian psittacine and equine hosts (82/88; 93.18%). However, this study also found novel hosts (Australian white ibis, King parrots, racing pigeon, bovine, and a wallaby) and demonstrated that strain diversity does exist in Australia. The discovery of a *C. psittaci* novel strain (ST306) in a novel host, the Western brush wallaby, is the first detection in a marsupial. Analysis of the results of this study applied a multidisciplinary approach regarding *Chlamydia* infections, equine infectious disease, ecology, and One Health. Recommendations include an update for the descriptive framework of *C. psittaci* disease and cell biology work to inform pathogenicity and complement molecular epidemiology.

## 1. Introduction

*Chlamydia psittaci* is traditionally regarded as an avian pathogen of global distribution with zoonotic potential [1]. *Chlamydia psittaci* infections are common in a variety of birds, including poultry, wild birds, and pet birds [1,2]. These infections are also found in humans, and to a lesser degree, domesticated livestock, such as sheep, cattle, horses, and pigs [2,3]. Spill-over transmission from the avian host to the non-avian host, via direct or indirect contact, is the hypothesised route of infection [1,4]. Compared to global studies [5,6,7], Australia has limited reporting in humans and domesticated livestock, such as sheep and cattle [4,8,9,10]. In contrast, *C. psittaci* has found a preferred host niche in the Australian psittacine birds [11,12] and Thoroughbred horses [13,14].

Whilst *C. psittaci* infections in Australian psittacine birds are not new [15], the first descriptions of *C. psittaci* in Thoroughbred horses in 2014 were considered novel and raised biosecurity concerns regarding a potential emerging infection [16]. *Chlamydia psittaci* infection in a pregnant horse resulted in late-term pregnancy loss and a novel infectious aerosol transmission of *C. psittaci* to humans handling infected equine foetal membranes [16,17]. Since then, *C. psittaci* has remained an annually reported cause of foal loss in the Hunter Valley region of Australia, a major Thoroughbred breeding center, as observed in a longitudinal study over four years from 2016 to 2020 [14,18]. A recent retrospective study has also provided evidence that this pathogen is not new in Australian horses and has been a cause of sporadic equine foal loss in Australia for over 30 years [19]. These studies highlight potential underreporting as *C. psittaci* was not previously considered in equine diagnostic panels. Addressing *C. psittaci* knowledge gaps is important considering the high individual value of the Thoroughbred foals and the zoonotic risk to farmworkers.

Globally, *C. psittaci* strains are genetically diverse, as evidenced by ongoing molecular characterisation by a chlamydial marker gene *omp*A [20], multilocus sequence typing (MLST) [21,22], and/or whole genome sequencing (WGS) [4,23]. The Chlamydiales MLST emerged as a “gold standard” methodology for molecular characterisation of strains, barcoding them as sequence types (STs). MLST of Australian equine samples typed to date have characterized a dominant strain, denoted ST24, in 53/55 samples (96.3%) [14,17,18,19,24] spanning three geographically separate regions in Australia (New South Wales [NSW], Victoria [Vic], and South Australia [SA]). This same strain has also been confirmed in a range of Australian psittacine birds, including little corella (*Cacatua sanguinea*), scaly breasted lorikeet (*Trichoglossus chlorolepidotus*), cockatiel (*Nymphicus hollandicus*), crimson rosella (*Platycercus elegans*), and the budgerigar (*Melopsittacus undulatus*) [12,25,26] from different Australian states (NSW, Vic), and importantly, humans from NSW [26]. Genomic studies have found that the Australian psittacine, horse, and human ST24 strains are highly clonal, with less than 80 single nucleotide polymorphisms (SNPs) differences [4,13] and that they cluster with other global virulent ST24 strains [22,23].

Despite the reported dominance of ST24, there is evidence that other STs can infect horses and birds in Australia. In a single case of equine foal loss in a northern region of Australia (Queensland [Qld]), a *C. psittaci* strain, denoted as ST27, was detected [13]. This “pigeon type” strain clustered with other closely related pigeon strains from Europe [22,27]. It has also been detected in a wild sulphur-crested cockatoo (*Cacatua galerita*) from a geographically different location (Vic.) [12]. Additionally, another closely related but distinct “pigeon type” strain (ST35) was detected from a spotted dove (*Spilopelia chinensis*) from yet another geographically different location (NSW) [13].

To date, no other *C. psittaci* STs, nor new hosts, have been identified in Australia, contrasting starkly with the known global host and genetic diversity [1,22,27]. These studies suggest that there may be a potential underreporting of *C. psittaci* host range and genetic diversity, perhaps masked by a dominant clonal ST24 strain, raising questions about the epidemiology of these infections in Australia. This study applies MLST and *omp*A genotyping to evaluate the widespread of the Australian ST24, determine the presence of other genetically distinct strains, and expand knowledge on the avian and mammalian host range. Furthermore, we applied and reviewed a multidisciplinary approach regarding *Chlamydia* infections, ecology, equine infectious disease, and One Health into discussion of this original work. This will provide future directions for *C. psittaci* research to prevent loss of Thoroughbred foals and potential zoonotic events.

## 2. Results

In this study, we applied *C. psittaci* MLST and *omp*A genotyping on 21 *C. psittaci*-positive samples to evaluate genetic diversity of these infections. The four new clinical samples from this study of an aborted foal (*Equus caballus*), a bovine (*Bos taurus*), and two corellas (*Cacatua sanguinea*) from cases of suspected chlamydiosis were confirmed *C. psittaci*-positive by a *C. psittaci*-specific qPCR assay (Appendix A). *Chlamydia psittaci* MLST and *omp*A genotyping were then applied to these four samples and 17 previously diagnosed *C. psittaci*-positive samples, selected from a range of Australian hosts (Appendix A). All 21 samples were successfully *omp*A genotyped; however, complete MLST was achieved for 19 samples. Complete MLST was not achieved for two samples (Ibis A and Cow UGT) due to low infection loads, and these were excluded from the MLST analysis.

### 2.1. Hosts, Infecting Strains Diversity and Clinical Signs

An evaluation of genetic diversity using MLST was performed on the 15 newly typed strains from this study and 73 previously characterised strains from all reported cases of *C. psittaci* cases in Australia (Appendix A). *Chlamydia psittaci*-specific MLST detected a single dominant clonal strain, ST24, in a total of 82/88 (93.18%) samples (Appendix A) from a total of 68/73 (93.15%) Australian animal and human hosts (Table 1). With regards to the strains in avian infections, all but one Australian psittacine (W234-12) strains were denoted ST24 (19/20, 95.00%). Clinical signs of emaciation, conjunctivitis, and dyspnoea, consistent with a diagnosis of psittacosis, were observed in 10 out of 19 (52.6%) Australian psittacine birds with ST24 infections. One of these infections resulted in zoonotic transmission (Adel_Rosella). In cases of foal loss, *C. psittaci* ST24 was detected in 57/59 samples (96.61%), except for one case in Queensland where ST27 strains were detected in two samples (2/59, 3.39%; Qld/H/pl; Qld/H/Tissues) (Appendix A). ST24 was detected in 44/45 (97.77%) of equine hosts (Table 1), with all 45 hosts presenting with clinical signs, and one of these events also included documented zoonotic transmission (Horse_Pl) (Appendix A). All cases of human psittacosis (*n* = 5 hosts) were previously described and were due to direct or indirect contact with presumably infected birds. All detected Australian human sequence types were ST24 (5/5; 100%), and severe clinical signs were observed in 4/5 (80%) hosts (Table 1).

The Australian non-psittacine birds (Racing_pigeon and NSW/Dove/tissue) displayed strain diversity (ST213 and ST35, respectively), although both had clinical signs consistent with psittacosis (Appendix A). A novel strain, ST306, was detected in a Western brush wallaby (*Macropus irma*) with the reported clinical signs of joint pathology (Table 1).

### 2.2. Phylogenetic Analyses of Detected C. psittaci Strains

The Australian strains were also phylogenetically compared with previous global 26 strains, selected for representative host and ST diversity (Appendix A). Newly typed strains from Australian king parrots (*Alisterus scapularis*), little corellas (*Cacatua sanguinea*), rosellas (*Platycercus elegans*), western rosellas (*Platycerus icterotis*), and horses, described in this study, resolved ST24 and clustered into well-supported globally distributed clonal ST24 clade (Figure 1A). The new *C. psittaci* strain ST306, detected in the wallaby, formed its own clade. The racing pigeon strain denoted ST213, was identical to a strain 394_T from a pigeon from Switzerland. This strain clustered in a broader “pigeon clade” with other closely related strains from Australian horses, a cockatoo and a dove, and global pigeon and mammalian hosts (Figure 1A).

Next, we evaluated genetic diversity using *omp*A genotyping on a total of 57 Australian strains. This consisted of 21 newly typed strains from this study, including samples from an Australian white ibis (*Threskiornis moluccus*) (Ibis_A) and a bovine (Cow_UGT), and 36 previously characterized Australian and global strains (Appendix A). The *omp*A genotyping revealed extended genetic diversity of *C. psittaci* in Australia; however, genotype A was the most common (Figure 1B). Almost all ST24 strains from this study had identical genotype A; however, variations of genotype A were also observed. The *omp*A genotypes of the Victorian psittacine strains (217, 207, 222), denoted A^a^, were identical to each other, but differed from genotype A by one non-synonymous SNP. The *omp*A from an Australian bovine, denoted A^b^, differed by six to seven SNPs from other genotype A sequences. Finally, the *omp*A sequences, denoted A^c^, from a seabird Fulmar (*Fulmarus glacialiscode*) and German horse strains, differed by one SNP between them and two from the *omp*A genotype.

The Australian racing pigeon had *omp*A genotype B, clustering with another identical genotype B, including that of NSW_Dove (Figure 1B). However, variations of genotype B (denoted B^a^) were also observed. The *omp*As from other global pigeon strains and an Australian horse (Qld_H_pl) strain were of these genetically diverse B^a^ genotypes. The *omp*A sequences detected in the wallaby and Australian white ibis were identical but differed by three SNPs from the reference VS225 *omp*A F, representing a novel *omp*A F^a^ genotype (Figure 1B).

For 64 strains, where we had paired *omp*A and MLST data, we also used Fastbaps with three levels of hierarchical clustering to further define genetic clusters based on MLST and *omp*A genotyping (Appendix A, Appendix A). The STs resolved into two primary clusters: monophyletic Cluster 1 (highlighted in orange) and polyphyletic Cluster 2 (highlighted in green), respectively (Appendix A). Cluster 1 consisted of “pigeon” STs, not forming any further clades. Cluster 2 further divided into three clades (2.2, 2.3, and 2.4) (Appendix A). Clades 2.2 and 2.3 consisted of genetically diverse STs, including the novel wallaby ST306, while Clade 2.4 consisted of ST24 and closely related ST218 (Figure 1a and Appendix A). Similarly, *omp*A alignment also resolved strains into two primary clusters; however, both clusters (Cluster 1 highlighted in orange, and Cluster 2 highlighted in green) were polyphyletic and further resolved into clades and subclades (Appendix A).

In this study, by utilizing MLST and *omp*A genotyping schemes, we revealed increasing genetic diversity, including the novel strain ST306 detected in a wallaby and novel *omp*A sequences detected in a cow and an ibis, and an expanded host range across all sampled states (with an exception of Northern Territory and Tasmania) in Australia, including a wallaby, bovine, psittacine (king parrot, Adelaide rosella, and western rosella), and non-psittacine (racing pigeon and Australian white ibis) birds (Appendix A).

## 3. Discussion

This study reveals an extended range of Australian animal hosts infected with clonal (ST24) *C. psittaci*, as well as novel (ST306) and previously described genetically diverse *C. psittaci* strains (ST213, ST35, ST27). Focusing on the Australian setting, ST24 is commonly found in psittacine, equine, and human infections [4,12,14,17,18,19]. Less commonly, there are isolated reports of the *C. psittaci* “pigeon” (ST27, ST35 and ST213) strains infecting a sulphur-crested cockatoo (*Cacatua galerita*), two columbids (*Columba livia domestica* and *Spilopelia chinensis*), and an equine (*Equus caballus*) [12,13]. To the best of our knowledge, this is the first evidence of a genetically novel strain (ST306/novel *omp*A) infecting a new native marsupial host, a western brush wallaby (*Macropus irma*). Furthermore, *omp*A genotyping revealed genetically diverse strains in a bovine Australian wallaby and white ibis, clustering with genotype A and F, respectively.

### 3.1. Uncovering C. psittaci Genetic Diversity

A comparison of global and Australian strains regarding diversity highlights the widespread of the ST24 in Australia. Most Australian studies, including the new molecular characterisation in this study, were performed on *C. psittaci-*positive samples from symptomatic and asymptomatic hosts [12,13,14,17,19,26]. However, global studies, often subjected to the same selection bias, still display a greater diversity in both strains and hosts [22]. Studies evaluating global *C. psittaci* genetic diversity identify at least 30 different STs and at least 15 different *omp*A genotypes, with an actively expanding range of hosts, including humans, domesticated livestock (sheep, cattle, pigs), and poultry (ducks, turkeys, chickens), and a wide range of free-range psittacine and non-psittacine birds [21,22]. Acknowledging the limitations in the Australian studies to date, are we underestimating the diversity of *C. psittaci* and host range in Australia?

Our findings suggest that there is likely to be an underestimation of diversity and host range in Australia. This might be the result of simply not testing a broad range of wildlife as well as domesticated animal species. *Chlamydia psittaci* has shown to be the most diversifying chlamydial species already a decade ago [21]. Indeed, to date, the number of novel STs of *C. psittaci* submitted to PubMLST is still growing, as well as the number of previously unrecognised host species. Furthermore, considering the increase in host species for *C. psittaci* [21], it is justifiable to suggest that diversity and host range is underestimated, not only in Australia but globally as well.

### 3.2. Roadmap to Understanding the Pathogenicity of Equine Abortogenic C. psittaci

Australian studies have focused on molecular descriptions of equine strains, which, on their own, cannot provide an informed understanding of the pathogenicity of each strain and the mechanism of equine foal loss. Indeed, any postulated potential altered pathogenicity/tropism in an emerging Australian equine strain must be interpreted with caution. In contrast, detailed cell biology studies are often performed overseas [2]. A recent European cell biology study has shown in vitro avian strains to be more pathogenic than mammalian strains regarding infection and zoonotic potential [28]. There is a lack of cell biology studies using Australian equine isolates, which may shed new light on mammalian pathogenicity.

We are still facing knowledge gaps on how *C. psittaci* is transmitted to the pregnant mare, how it colonises the placenta, and how pathological processes in the foetal membranes/organs are induced. If the infectious agent (chlamydial elementary bodies [EBs]) is orally ingested from pastures contaminated with faecal droppings containing *C. psittaci*, information about the density of the bird population shedding *C. psittaci* is of importance. While psittacine birds might play a major role in Australia, pigeons might be considered as a potential infection source in European countries. Moreover, the case numbers of equine foetal loss in the respective geographical regions are closely related to the type and density of horse farming and breeding. Mixed infection of *C. psittaci* and other abortigenic agents (i.e., Equine Herpes Virus 1 [EHV-1]) have been reported [22] but it remains unclear whether one of the agents (or a combination of both) induced the abortion. Specific laboratory methods, such as immunohistochemistry and in-situ hybridisation, are helpful to demonstrate the agent in associated tissue lesions. Further characterisation of histopathological lesions and inflammatory processes would help us to understand this ongoing pathogenesis, as learnt from field and experimental infection studies of ovine enzootic abortion caused by *C. abortus*, although the latter might not be justified in the equine host.

### 3.3. Are We Missing Other Hosts?

Other major equine breeding centres worldwide have not reported chlamydial foal loss despite globally reported *C. psittaci* prevalence in birds of between 1–5% [1,11]. Should we be exploring host diversity closer to home for an answer? An area of future research could involve Australian fauna that shares a common habitat with livestock. The detection of *C. psittaci* in a wallaby joint in this study is the first detection of *C. psittaci* in a marsupial. However, the severe impact of *C. pecorum* infections on Australian koalas, resulting in reproductive, urinary, and ocular disease, is well known [8,29]. Earlier reports of *C. psittaci* in koalas have since been revised, and to date, there remains no evidence of *C. psittaci* in koalas, nor any other marsupial. Considering the close contact koalas have with Australian birds in the shared arboreal habitat, and the *C. psittaci* preferred niche of the respiratory and reproductive tract, should *C. psittaci* be considered on a diagnostic panel for koala disease? Furthermore, should other marsupials be investigated considering the shared landscape?

It is becoming clear that several species of *Chlamydia* that had previously not been considered to be present in hosts, such as horses, marsupials, and others, do indeed infect these hosts. Part of the issue is that, previously, we were not considering *C. psittaci* as a potential cause and did not routinely test specifically for this species. If we test more hosts, such as Australian marsupials, for *C. psittaci*, might we expect to find more? Given the opportunity for transmission between hosts due to co-location, this may well be the case.

Although the role of Chlamydia in disease is well known in koalas and other Australian marsupials, *C. psittaci* is not actively investigated for in marsupials by veterinary diagnosticians, and its presence and impact are not known. The detection of *C. psittaci* from the joint of a wallaby in this study is significant and suggests a broader host range and possible pathogenicity in marsupials than we currently understand. We sporadically observe inflammatory lesions in koalas and other marsupials, for which aetiology is not yet determined. Targeted investigation of *C. psittaci* in Australian marsupials would be of great benefit; this study suggests the host range, and thus conservation and zoonotic risk impacts may be broader than we presently appreciate. 

Australian flying foxes (*Pteropus* spp. fruit bats) are known reservoirs for zoonotic viruses, including Hendra virus, Australian bat lyssavirus (ABLV), and Menangle virus [30,31,32]. Hendra virus spill-over transmission from flying foxes to horses occurs annually, including recently in the Upper Hunter Valley of NSW, a major Thoroughbred breeding centre [33,34,35]. Overseas studies have observed a prevalence of 31.4% of *Chlamydiales* in free-ranging and captive bats from a range of families in the Yinpterochiroptera and Yangochiroptera bat suborders [36], but no data currently exist on chlamydial infections in Australian bats. Understanding what is happening in Australia at the wildlife/equine/human interface provides knowledge about the emerging zoonotic disease.

As nectar feeders and pollinators, Australian flying foxes play critical ecosystem roles as they move nomadically across the landscape, feeding on flowering native trees. When nectar is scarce, for example, as a result of climate cycles and loss of habitat due to land clearing, flying foxes will seek out alternative food sources, including fruit trees in urban and agricultural areas. Subsequently, increased contact with horses and other domestic animals may amplify the risk of pathogen spill-over. Further investigations into the infectious agents circulating endemically in Australian bats, as well as systematic pathogen discovery efforts following unusual deaths in domestic animals and livestock, is required to detect the extend of pathogen spill over from bats. Sustainable One Health solutions, such as habitat restoration, present the most promising opportunities for preventing spill-over [37].

### 3.4. Diagnostic Directions for C. psittaci Equine Abortion and Zoonosis

A diagnosis of *C. psittaci* in the equine is challenging [14,38]. Whilst high loads are noted in clinical cases of both avian psittacosis and equine foal loss, the significance of detection of low levels in a healthy host remains unknown [18]. Importantly, early detection for intervention in the pregnant mare is still lacking. Considering that Australia’s major Thoroughbred breeding centre in the Upper Hunter Valley is in a potentially endemic ST24 region, what risk does the emerging strain diversity pose for an outbreak of foal loss and human psittacosis?

There are now multiple reports implicating *C. psittaci* as a cause of abortion in horses. This is an emerging pathogen that must be included in routine microbiological diagnostic testing of all equine abortions. It is not clear whether there is a clearly delineated geographical distribution of *C. psittaci* abortion, but heightened awareness and active surveillance in the future will help to address this concern. The development of excellent molecular diagnostic tests will enable the rapid diagnosis of this organism in clinical samples, which in turn will improve surveillance for this organism. There is a serious zoonotic potential associated with *C. psittaci* and so equine veterinarians and staff on equine breeding farms must take more precautions when dealing with the aborted equine conceptus and the affected mare.

Equine parturition can now be considered as a time of higher risk for transmission of *C. psittaci* to farmworkers [16,39]. Compared to global reporting, there is potential underreporting in Australia, with less than 50 notified cases of psittacosis per annum in the past decade reported in the Australian National Notifiable Diseases Surveillances (ANNDS) [40]. To date, some regions are considered endemic, and this study has identified the upper Hunter Valley as another potential region. The recent advent of *C. psittaci* rapid diagnostic testing in horses [41] shows promise for farm management strategies of *C. psittaci* and highlights the importance of front-line workers for zoonotic disease. As the world moves towards the more informed One Health approach to disease, a revision of the term psittacosis to chlamydiosis is suggested. This study has provided evidence of an expansion of a host range with zoonotic potential and the term psittacosis in the lexicon may be misleading and limit potential host investigations. 

The trans-disciplinary benefits of a One Health approach to investigate the epidemiology of *C. psittaci* will improve our understanding and management of risk, not just for this disease agent but for zoonoses more broadly. There are additional benefits to landscape ecosystem health [42] and to local capacity to manage threats at the animal–human–wildlife–environment interface [43].

In summary, the molecular analyses of *C. psittaci* have shown the widespread of a dominant strain (ST24) in Australia with the serious consequences of morbidity in people and animals, and mortality in equines. This work has identified a potentially endemic region within Australia and an expanded host range for *C. psittaci*, including a novel marsupial, an expanded bird range, and a bovine. Future directions include a recommendation to update the descriptive framework of *C. psittaci* disease, implement ongoing surveillance and strain identification for potential outbreak intervention, and include cell biology work to inform pathogenicity.

## 4. Materials and Methods

### 4.1. Sample Collection, DNA Extraction and C. psittaci Screening

In this study, molecular characterisation of *C. psittaci* was performed on a combined set of 4 new and 17 existing samples which were opportunistically collected from a range of sources and hosts across Australia. The four new samples from this study were collected from a new 2020 case of *C. psittaci*-suspected aborted foal from the Hunter Valley region, NSW (pooled foetal swab; *n* = 1), a clinically affected bovine from Western Australia (WA) (urogenital tract (UGT; *n* = 1), and two symptomatic dead little corellas from a wildlife hospital in Qld (pooled eye/choana and cloaca swabs; *n* = 2). All dry and individually capped swab samples were collected as a part of routine veterinary diagnostic investigations and stored at −20 °C until processing. Swabs were then processed by adding 300 µL of sterile TE (Tris-EDTA) buffer, vortexing for one min, and heating at 95 °C for 10 min. Heat lysis was used to improve the host cell lysis method and release (strict intracellular) chlamydial elementary bodies into solution. DNA extraction was performed using the QIAmp DNA blood and tissue kit (Qiagen, Chadstone, VIC, Australia), as per the manufacturer’s instructions. Extracted DNA was eluted in 100 µl of supplied AE buffer (Qiagen) and stored at −20 °C until further analysis. The DNA was screened in duplicate for *C. psittaci* with the *C. psittaci*-specific qPCRs targeting a 263 bp fragment of the ORF 607 gene with a Ct result of <33 being deemed positive [8].

This sample set was supplemented with the addition of 16 previously tested *C. psittaci*-positive DNA samples from a range of hosts across Australia, including equine (*n* = 3), avian (total *n* = 12 [psittacine *n* = 11 and columbid *n* = 1]), and marsupial (*n* = 1) hosts, which were generously supplied by collaborators from veterinary diagnostic laboratories across Australia. These samples were initially screened by the respective veterinary diagnostic reference of *C. psittaci*-specific assays, using the CPS100 and CPS101 primers which targeted the 16S rRNA gene/16S–23S rRNA space [44] or 76bp fragment of the *C. psittaci* outer membrane protein A (*omp*A) gene [45]. We also used a *C. psittaci*-positive sample (Ibis_A) from an asymptomatic Australian white ibis (*Threskiornis moluccus*) from the same region as the Thoroughbred horses NSW (faeces; *n* = 1), from a recent study [18]. Prior to further analyses, all samples were internally confirmed for *C. psittaci* using the same *C. psittaci*-specific qPCR assays as outlined above for new samples [17]. The final catalogue of samples for molecular characterisation included 21 samples (4 detected in newly collected samples and 17 in previously screened samples). Details are available for all samples in this study regarding host, anatomical site, clinical manifestation, location, and date (Appendix A). Trained and experienced veterinary teams collected all swabs, and ethical approval for using these swabs was granted by the University of the Sunshine Coast (ANE 1939/ ANE2057).

### 4.2. Genotyping

*C. psittaci* genotyping was applied on a total of 21 samples from this study, using *C. psittaci*-specific MLST [46] and/or full-length major outer membrane (*omp*A) gene sequencing [20] as previously described. 

Conventional PCRs were performed in a total of 35 µL reaction volumes, including 17.5 µL of the 2X Amplitaq Gold master mix (Thermo Scientific, Scoresby, VIC, Australia), 1 µL of 0.3 µM primer (using ten µM working concentration), 12.5 µL of MilliQ water, and 4 µl of DNA template with cycling profiles as previously described [45]. PCR products were visualised on 1.5% agarose gel stained with SyberSafe following gel electrophoresis for 30 min on 100V. Amplicons were then sent for purification and bidirectional Sanger sequencing (Macrogen, Seoul, Korea). Chromatograms were examined for quality and analysed in Geneious Prime 2020.2.4 [47]. Chromatograms of low quality, evaluated based on Phred scores and lengths shorter than the reference amplicons, were discarded, and sequencing was repeated. The MLST sequences were compared with the Chlamydiales MLST database [48] to determine and/or assign novel alleles and sequence types (ST). The newly generated *omp*A sequences were subjected to BLASTn analysis [49]. MLST and *omp*As sequences generated in this study were deposited in the *Chlamydiales* MLST database and GenBank with accession numbers MZ207919–MZ207938 and MZ298912.

### 4.3. Phylogenetic and Cluster Analyses

The 21 *omp*A and 19 MLST sequences from this study were then aligned using ClustalOmega (as implemented in Geneiois Prime 2020.2.4) to other representative, publicly available *omp*A and/or MLST sequences. Metadata for these samples (including genotype, hosts, geographical locations, year of isolation, clinical manifestations, and other) are outlined in Appendix A. The midpoint rooted maximum likelihood tree was constructed using the 3098 bp alignment of concatenated MLST from a total of 74 strains and 1050 bp *omp*A alignment from *omp*A sequences from a total of 66 strains with FastTree using the GTR + G nucleotide substitution model (as implemented in Geneious Prime) with a strain M56 used as an outlier. 

Paired *omp*A and MLST sequences were available for 64 strains analyzed in this study (Appendix A). To further assess *C. psittaci* population structure and cluster strains into genetically close groups (phylogroups), a paired *omp*A and MLST alignments from 64 strains were analyzed using the R package FASTbaps [50] to partition the data into groups using Bayesian clustering (using the symmetric method for prior optimization and three levels of clustering) (Appendix A, Appendix A).

## Figures and Tables

**Figure 1 pathogens-10-01015-f001:**
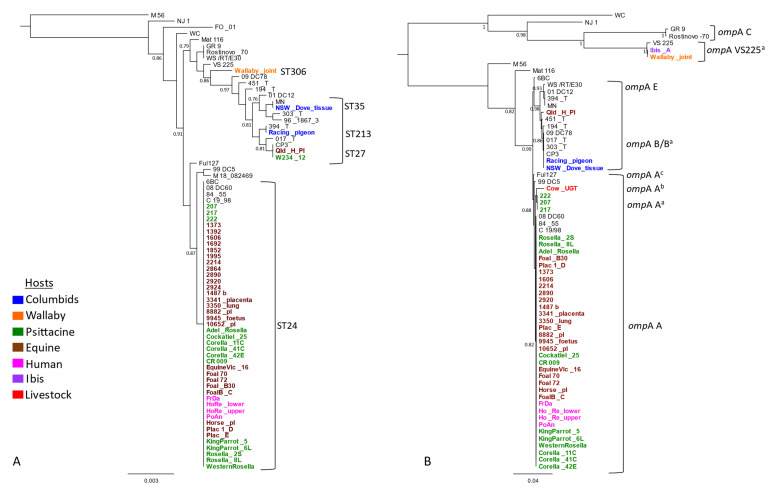
Genetic diversity and phylogenetic analyses of *C. psittaci* from this study. The mid-point rooted phylogenetic analyses of (**A**) the concatenated 3098 bp *C. psittaci* STs alignment, and (**B**) full length (1050 bp) *omp*A genotypes alignment. Support values of >0.80 are displayed on the tree nodes, and the scale bar represents the number of substitutions per site. The hosts are indicated as per colours in the legend. Australian isolates are denoted in bold and in colours as per host (outlined in the legend), while global isolates are denoted in normal black font.

**Table 1 pathogens-10-01015-t001:** Hosts and their infecting strain diversity with noted presence or absence of clinical signs of chlamydiosis in Australia. The clinical signs are described in Appendix A in detail for each host and include: pneumonia in humans; wasting, dyspnoea, and conjunctivitis in birds; and reproductive loss in horses.

Hosts	Avian		Equine	Human	Wallaby
	Psittacine	Columbids			
No. of ST24-positive hosts	19/20 (95%)	0/2 (0%)	44/45 (97.8%)	5/5 (100%)	0/1 (0%)
Total No. of hosts with clinical signs in ST24 infections	11/19 (57.9%)	-	44/44 (100%)	4/5 (80%)	-
No. of other STs-positive hosts	1/20 (5%)	2/2(100%)	1/45 (2.2%)	0/6 (0%)	1/1 (100%)
Total No. of hosts with clinical signs in other STs infections	0/1 (0%)	2/2(100%)	1/1 (100%)	-	1/1 (100%)

## Data Availability

The MLST and *omp*A sequences from this study were deposited in the Chlamydiales PubMLST database and Genbank.

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
