# Peer review of "Chlamydia Psittaci ST24: Clonal Strains of One Health Importance Dominate in Australian Horse, Bird and Human Infections"

_pathogens, 2021, doi:10.3390/pathogens10081015_

Round 1

Reviewer 1 Report

The authors have analysed different host species (4) for C.psittaci and characterised the strains. To their surprise this has been found in more host than initially expected and a new strain was characterised. Overall the results are poorly described and do not guide the reader. The result section should be more detailed 

Unclear sentence (typos?) line 29, 45 

Typos line 145 

‘ success of ST24’ is mentioned through the text, what measure success for a STD? 

Figure 1 is very confusing, the arrow system triggers the reader to think the infection propagate through all the species, for example for the first one the eye quickly conclude that it infects two type of birds and then a horse and then human. Then the arrow have a complex colour code, which seem to reflect more accurately what they authors communicate. So why the arrow. ‘Postulated spill-over' postulated on the basis of what? Is that bringing something? Do we need the postulat? 

Figure one legend is about number and then there are ratio. What are those two numbers? If the divisor is the total population, why the discrepancy between the number of case and the number of hosts with symptoms? 

What are the symptoms? 

Paragraph 2.2, why the use of these two methods to produce to different trees? What are you trying to achieve? 

Line 174 ‘an extended range of Australian animal hosts infected with clonal...’ four species and 5o+ samples can not really qualify for extended 

Line 175 ‘ll as novel (ST306)’ this is barley mentioned in the result. Novel on which basis? 

A couple of time it is mentioned that a greater diversity is expect, it would be good to have more details. How many more species? How many samples in total? 

The discussion represent half of the article. I do think that more effort have to be made to have a better description of the data which is missing here 

The provided supplementary information are not mentioned in the text, we do not know in which context they should be read 

Author Response

Reviewer 1

The authors have analysed different host species (4) for C.psittaci and characterised the strains. To their surprise this has been found in more host than initially expected and a new strain was characterised. Overall the results are poorly described and do not guide the reader. The result section should be more detailed.

Authors reply: We thank you for your insightful comments. We have addressed them to the best of our ability, and we have put efforts to improve results section.

Unclear sentence (typos?) line 29, 45, Typos line 145 

Authors reply: Thank you for this comment. We have corrected the sentences to read.

Line 29: “and demonstrated that strain diversity does exist in Australia”

Line 45: Australia has limited reporting in humans and domesticated livestock,… “

Line 145:” The racing pigeon strain denoted ST213, was identical to a strain 394_T from a pigeon from Switzerland. This strain clustered in a broader "pigeon clade" with other closely related strains from Australian horses, a cockatoo and dove, and global pigeons and mammalian hosts (Figure 2a).”

‘ success of ST24’ is mentioned through the text, what measure success for a STD? 

Authors reply: Chlamydia psittaci does not cause STD (sexually transmitted diseases), it is a (primarily) respiratory pathogen. But asymptomatic shedding is also noted. Psittacosis (C. psittaci chlamydiosis) can also result in systemic disease. In here, we refer to this genotype ST24 as the most successful type strains due to its pathogenic potential to cause respiratory systemic diseases and equine abortions, as well as to easily spill-over to humans (causing zoonotic disease).

Figure 1 is very confusing, the arrow system triggers the reader to think the infection propagate through all the species, for example for the first one the eye quickly conclude that it infects two type of birds and then a horse and then human. Then the arrow have a complex colour code, which seem to reflect more accurately what they authors communicate. So why the arrow. ‘Postulated spill-over' postulated on the basis of what? Is that bringing something? Do we need the postulat? 

Figure one legend is about number and then there are ratio. What are those two numbers? If the divisor is the total population, why the discrepancy between the number of case and the number of hosts with symptoms? What are the symptoms? 

Authors reply: Thank you for these comments, we have adapted this figure to denote only number of hosts infected with ST24 and other STs, and number of hosts with clinical signs in these infections, and we provide detailed legend. We also simplified the arrows and added icons to better denote hosts. The postulated spillover is based on molecular evidence to date, and is important to understand reservoirs for this pathogen.

Please see the new figure.

Figure 1. Hosts, and infecting strain diversity with noted presence or absence of clinical signs of chlamydiosis in Australia. The clinical signs are described in Table S1 in detail for each host and include: pneumonia in humans; wasting, dyspnoea, and conjunctivitis in birds; and reproductive loss in horses. The arrows and icons denote postulated spillover of C. psittaci infections (in a direction from-to). *: Postulated spillover of infections is based on molecular evidence to date. The psittacine hosts are represented with green arrows, columbids with a blue arrow, and equine with a brown arrow. The question mark denotes unknown spillover.

Paragraph 2.2, why the use of these two methods to produce to different trees? What are you trying to achieve? 

Authors reply: The two trees are phylogenies based on concatenated MLST alignment (in Panel A) and ompA gene alignment (in B). MLST phylogeny is highly congruent with whole genome phylogeny, while ompA (as the major chlamydial antigen) is a polymorphic gene and adds to further molecular characterisation of the strains. We have outlined in methods that two different targets were used in phylogenetic analyses to see genetic diversity using 2 different molecular targets and to denote clonality of the ST24 clade (as all ST24 strains also had identical ompA genotype A).

Line 174 ‘an extended range of Australian animal hosts infected with clonal...’ four species and 5o+ samples can not really qualify for extended 

Authors reply: To date, in Australia C. psittaci has only been described in human, psittacine only 2 pigeons, and equine hosts. Detecting C. psittaci in a native marsupial wallaby, extended bird range: King Parrot, Australian White Ibis, Little Corellas, and a cow reveals an expanded host range for Australia. We do agree with the reviewer that this is could be considered restrict host range elsewhere. In Australia, the attention to this pathogen was brought by C. psittaci equine epizootic in 2016 and other from 2016 – 2019 equine cases (where we had the most numerous C. psitaci positive samples (n=55)). Other studies detected few positive samples (often less than 10). Hence, although we were able to collate only 21 samples for this study, we wish to point that these samples are often are truly opportunistic but represent valuable catalogue to decipher epidemiology and reservoirs for these infections.

Line 175 ‘ll as novel (ST306)’ this is barley mentioned in the result. Novel on which basis? 

Authors reply: The ST306 wallaby strain is novel based on several findings: firstly, this is the first time that C. psittaci is detected in a novel native Australian marsupial host, the wallaby; second: infecting strain is characterised by novel ST as well novel ompA genotype (molecularly characterising a novel genetically distinct C. psittaci strain). We do reiterate this in results.

A couple of time it is mentioned that a greater diversity is expect, it would be good to have more details. How many more species? How many samples in total? 

Authors reply: In this study we analysed 21 new samples to all previous 73 samples from Australia. As stated above, there are not many molecular studies on C. psittaci. In the results section we have added summary to reflect this.

Line xxx: ‘ In this study, utilising MLST and ompA genotyping schemes, we revealed increasing genetic diversity, including the novel strain ST306 detected in a wallaby and novel ompA sequences detected in a cow and an ibis, and an expanded host range across all sampled states (with an exception of Northern Territory and Tasmania) in Australia including a wallaby, bovine, psittacine (king parrot, Adelaide rosella and western rosella) and non-psittacine (racing pigeon and Australian white ibis) birds (Supplementary Figure 2).”

The discussion represent half of the article. I do think that more effort have to be made to have a better description of the data which is missing here 

 Authors reply: We have streamlined the discussion and improved our results for more clarity.

The provided supplementary information are not mentioned in the text, we do not know in which context they should be read 

Authors reply: Thank you for pointing this out, you are correct. The Supplementary data should have been clearer. We have now updated this better in the text. Also, we have simply omitted results pertaining to Supplementary Figure 1 and Table S2.

Reviewer 2 Report

In this manuccript by Anstey et al., the authors report genotypes of Chlamydia psittaci among different hosts in Australia. They apply MLST and OmpA genotyping to investigate the genetic diversity of C. psittaci. They conclude that MLST type ST24 dominates among different host in Australia while there are also other ST types that can cause infection in hosts (including novel host reported here). They also include discussion with experts from different field into their paper. 

Major comments

  1. Sample information: There is ambiguity in description of samples used in this study. Example includes lines 101-105 (line 104 stated 15 samples from this study and 4 from previous studies while it seems contradictory to that stated in the previous line 101) , line 107 (Why only 15 MLST types were used for analysis when 19 samples were successfully MLST typed?). I would suggest the authors have a separate section in methods for full description of this.
  2. What are the sources of other samples used in this study (73 or 74 samples from previous studies)? 
  3. Samples size seems low for this study (only 21 samples from this study while other samples used in analysis are from previous study).

Minor comments

  1. Few Typos or grammatical errors. Example: line 29 in the Abstract states "strain diversity in does exist..........". line 45 states ",Australia has limited reporting in Australia in......."
  2. The genus name for the pathogen has been used in full where it could be just written as C. psittaci. example line 101, 110. 

Author Response

Reviewer 2

In this manuccript by Anstey et al., the authors report genotypes of Chlamydia psittaci among different hosts in Australia. They apply MLST and OmpA genotyping to investigate the genetic diversity of C. psittaci. They conclude that MLST type ST24 dominates among different host in Australia while there are also other ST types that can cause infection in hosts (including novel host reported here). They also include discussion with experts from different field into their paper. 

Authors reply: We thank you for your insightful comments. We have addressed them to the best of our ability.

Major comments

  1. Sample information: There is ambiguity in description of samples used in this study. Example includes lines 101-105 (line 104 stated 15 samples from this study and 4 from previous studies while it seems contradictory to that stated in the previous line 101) , line 107 (Why only 15 MLST types were used for analysis when 19 samples were successfully MLST typed?). I would suggest the authors have a separate section in methods for full description of this.

 Authors reply: To clarify, in total we had 21 samples for typing (17 samples previously diagnosed as C. psittaci in diagnostic laboratories (but never molecularly characterised) and, 4 new samples that we diagnosed and included in the study). ompA was achieved for all, but MLST only for 19 (2 samples were low loads and could not achieve all 7 MLST gene PCRs). We have now clarified this in results and methods.

Results: Lines 97 – 107: “In this study, we applied C. psittaci MLST and ompA genotyping on 21 C. psittaci positive samples to evaluate genetic diversity of these infections. The four new clinical samples from this study of an aborted foal (Equus caballus), a bovine (Bos taurus), and two corellas (Cacatua sanguinea) from cases of suspected chlamydiosis were confirmed C. psittaci positive by C. psittaci-specific qPCR assay (Table S1). Chlamydia psittaci MLST and ompA genotyping were then applied to these four samples and 17 previously diagnosed C. psittaci positive samples, selected from a range of Australian hosts (Table S1). All 21 samples were successfully ompA genotyped, however complete MLST was achieved for 19 samples. Complete MLST was not achieved for two samples (Ibis A and Cow UGT) due to low infection loads, and these were excluded from the MLST analysis.”

Methods:

Line 335:” In this study, molecular characterisation of C. psittaci was performed on a combined set of four new and existing 17 samples opportunistically collected from a range of sources and hosts across Australia.

The four new samples …”

Lines 348 - onwards describe these 17 previously tested samples.

  1. What are the sources of other samples used in this study (73 or 74 samples from previous studies)? 

Authors reply: We do point the reader to Table S1 (which includes metadata for all samples in this study) on Line 361, however we agree that this is rather lost in text. We added following after mentioning publicly available samples:

Line 385: “ The 21 ompA and 19 MLST sequences from this study were then aligned using ClustalOmega (as implemented in Geneiois Prime 2020.2.4) to other representative publicly available ompA and/or MLST sequences. Metadata for these samples (including genotype, hosts, geographical locations, year of isolation, clinical manifestations and other) are outlined in Table S1.”

  1. Samples size seems low for this study (only 21 samples from this study while other samples used in analysis are from previous study).

Authors reply: Yes, that is correct and we do agree with the reviewer. In Australia, the attention to this pathogen was brought by C. psittaci equine epizootic in 2016 and other equine cases up to 2020 (where we had the most numerous C. psitaci positive samples (n=55)). Other studies also detected few positive samples (often less than 10). Hence, although we were able to collate only 21 samples for this study, we wish to point that these samples represent valuable catalogue to decipher epidemiology and reservoirs for these infections. Now we have a basis to extend our screening for C. psittaci in a variety of hosts.

Minor comments

  1. Few Typos or grammatical errors. Example: line 29 in the Abstract states "strain diversity in does exist..........". line 45 states ",Australia has limited reporting in Australia in......."

Authors reply: Thank you for this comment. We have corrected the sentences to read.

Line 29: “and demonstrated that strain diversity does exist in Australia”

Line 45: Australia has limited reporting in humans and domesticated livestock,… “

  1. The genus name for the pathogen has been used in full where it could be just written as C. psittaci. example line 101, 110. 

Authors reply: Thank you for this comment. We were advised to spell out the genus/species in full at the beginning of a sentence.

Reviewer 3 Report

In general the manuscript is a good.

The topics is really hot. 

I recommend a few modification.

Double check the English.

Please describe the abbreviations when you use them for the first time

In the discussion part you need to compare with other studies from your country or surrounding  region. 

I  think the conclusions could be streamlined to become more precise and clear!

The figure 2 is extremely hard to read, maximize the writing. 

The paragraph 3 row  between 212- 229  is not clear can be rewrite in more clear format.

Thank you again for the opportunity to review this interesting manuscript. 

Author Response

Reviewer 3

In general the manuscript is a good. The topics is really hot. 

Authors reply: We thank you for your insightful comments. We have addressed them to the best of our ability.

I recommend a few modification. Double check the English.

Please describe the abbreviations when you use them for the first time.

Authors reply: The grammar, syntax and spelling were rechecked throughout the manuscript.

In the discussion part you need to compare with other studies from your country or surrounding region. 

Authors reply: Thank you for this comment. We have amended the comparison to include more details in regards to hosts and STs.

See revised Lines 218 – 220:” Studies evaluating global C. psittaci genetic diversity identify at least 30 different STs and at least 15 different ompA genotypes, with actively expanding range of hosts, including humans, domesticated livestock (sheep, cattle, pigs) and poultry (ducks, turkeys, chickens), and wide range of free-range psittacine and non-psittacine birds [21,22].”

I think the conclusions could be streamlined to become more precise and clear!

Authors reply: We amended the conclusions as advised.

Lines: “In summary, the molecular analyses of C. psittaci have shown the success of a dominant strain (ST24) in Australia with the serious consequences of morbidity in people and animals, and mortality in equines. This work has identified a potentially endemic region within Australia and an expanded host range for C. psittaci, including a novel marsupial, an expanded bird range and a bovine.  Future directions include a recommendation to update the descriptive framework of C. psittaci disease, implement ongoing surveillance and strain identification for potential outbreak intervention and include cell biology work to inform pathogenicity.”

The figure 2 is extremely hard to read, maximize the writing. 

Authors reply: The figure 2 is supplied separately, for review purposes is included in the text, where writing appears smaller.  Furthermore, we have decreased the amount of white space (as best as possible) and all global sequences are now in black to better highlight sequences from Australian hosts. Due to many sequences included in the tree, we used the most optimum font to display all strain names (e.g larger fonts squash sequence names)

The paragraph 3 row between 212- 229  is not clear can be rewrite in more clear format.

Authors reply: We have corrected as recommended.

Thank you again for the opportunity to review this interesting manuscript. 

Authors reply: Thank you again for such positive feedback on our manuscript.

Reviewer 4 Report

The manuscript is well written, but the discussion style should be changed, this is not a commentary.

Author Response

Reviewer 4

The manuscript is well written, but the discussion style should be changed, this is not a commentary.

 Overall, the paper is important and well written, there are a few questions and suggestions that will make the paper better. These samples are hard to come by and so this work is very relevant. Understanding what type of chlamydia is being transmitted in animals that might be reservoirs for this pathogen.

Authors reply: We thank you for your positive feedback and insightful comments. We have addressed them to the best of our ability.

Abstract

Introduction

Typo- rewrite, Line 45

Authors reply: Thank you for this comment. We have corrected the sentence to read.

Line 45: Australia has limited reporting in humans and domesticated livestock, … “

Methods

I would like to find out if it’s normal to heat the swabs before DNA extraction, it would be nice to justify why that procedure was used.

Authors reply: Heat lysis of the swab suspensions was performed for 2 reasons: for biosecurity purposes to inactive (other) infectious agents that may be present in the tissue; and, Heat lysis was used to improve the host cell lysis and release (strict intracellular) chlamydial elementary bodies into solution prior to DNA extraction. We use this method commonly in our lab.

We have added tis into the methods: Line 342:” Heat lysis was used to improve the host cell lysis and release (strict intracellular) chlamydial elementary bodies into solution.”

Results

Rewrite Line 98-101, make it clearer and easier to understand. Is there a discrepancy in table one regarding to the number of human samples that were positive? I found it difficult to square the number of humans tested in figure 1 and S1.

Authors reply: We have amended these sentences for clarity.

Results: Lines 97 – 107: “In this study, we applied C. psittaci MLST and ompA genotyping on 21 C. psittaci positive samples to evaluate genetic diversity of these infections. The four new clinical samples from this study of an aborted foal (Equus caballus), a bovine (Bos taurus), and two corellas (Cacatua sanguinea) from cases of suspected chlamydiosis were confirmed C. psittaci positive by C. psittaci-specific qPCR assay (Table S1). Chlamydia psittaci MLST and ompA genotyping were then applied to these four samples and 17 previously diagnosed C. psittaci positive samples, selected from a range of Australian hosts (Table S1). All 21 samples were successfully ompA genotyped, however complete MLST was achieved for 19 samples. Complete MLST was not achieved for two samples (Ibis A and Cow UGT) due to low infections loads, and these were excluded from the MLST analysis.”

Is there a discrepancy in table one regarding to the number of human samples that were positive? I found it difficult to square the number of humans tested in figure 1 and S1.

Authors reply: We have amended the Figure 1 for clarity.

The discrepancy in the table appeared as one of the rows discuses Number of samples, and other Number of hosts. Now, we only refer to number of hosts with ST24 and other STs infections, and number of hosts with clinical signs in these. We also added icons to better denote hosts.

Please see the new figure.

Figure 1. Hosts, and infecting strain diversity with noted presence or absence of clinical signs of chlamydiosis in Australia. The clinical signs are described in Table S1 in detail for each host and include: pneumonia in humans; wasting, dyspnoea, and conjunctivitis in birds; and reproductive loss in horses. The arrows and icons denote postulated spillover of C. psittaci infections (in a direction from-to). *: Postulated spillover of infections is based on molecular evidence to date. The psittacine hosts are represented with green arrows, columbids with a blue arrow, and equine with a brown arrow. The question mark denotes unknown spillover.

We believe you also additionally refer to Figure S2: Figure S2 is just overall schematic of hosts where we detected C. psittaci, locations across Australia and detected STs. We amended Figure S2 legend: “Figure S2: Schematic representation of various hosts for C. psittaci and MLST strain identity from Australia”.

Discussion

The discussion is well written; however, I do not see the need for having it written in a commentary style. I think the authors should rewrite the discussion removing the names of these experts. Their names are already in the authors list. That way the discussion will follow the normal style of written discussion. It should be understood that the opinion and answers given to the questions raised in this research is good, but the names should be removed from the discussion.

Authors reply: We have amended the discussion as suggested – the names are now removed and now follows the normal style of written discussion. Please see tracked changes in the discussion.

Round 2

Reviewer 1 Report

Some minor typos (line225 for example)

I remain no certain of the need of half of the figure 1 with the horses, people and arrows now with a huge question mark. A figure to explain a postulate which is described in the text is bringing to much emphasis on something which is not demonstrated and in this very paper.

I apology for the review referring to the strain as responsible of the STD while it is a respiratory infection. Nevertheless, the definition of success being the accomplishment of an aim or purpose, I was trying to communicate that in the context of this article this consist an anthropomorphism... unless it has been proven that Chlamydiae have purposes.

If I can accept to have the figure 1 left as it is, I would insist that this type of lab slang (which we all use no judgment here) remains in labs

Author Response

Authors response: We thank you for this further feedback, we have amended as advised.

Some minor typos (line225 for example)

Authors response: The manuscript was checked again, and these have been rectified.

I remain no certain of the need of half of the figure 1 with the horses, people and arrows now with a huge question mark. A figure to explain a postulate which is described in the text is bringing to much emphasis on something which is not demonstrated and in this very paper.

Authors response: We have removed this part of the figure (now the data are presented in a Table). Please see updated Table 1.

I apology for the review referring to the strain as responsible of the STD while it is a respiratory infection. Nevertheless, the definition of success being the accomplishment of an aim or purpose, I was trying to communicate that in the context of this article this consist an anthropomorphism... unless it has been proven that Chlamydiae have purposes.

Authors response: Thank you for this comment. Throughout the manuscript (where success was used, total of 5 times), we now exchanged “success” for “dominance” and/or “widespread”.

If I can accept to have the figure 1 left as it is, I would insist that this type of lab slang (which we all use no judgment here) remains in labs

Authors response: Please see above. The figure has been exchanged for a Table.